# Adipo-Modulation by Turmeric Bioactive Phenolic Components: From Curcuma Plant to Effects

**DOI:** 10.3390/ijms26146880

**Published:** 2025-07-17

**Authors:** Cristina Doriana Marina, Daniela Puscasiu, Corina Flangea, Tania Vlad, Adinela Cimporescu, Roxana Popescu, Aurica Elisabeta Moatar, Daliborca Cristina Vlad

**Affiliations:** 1Department of Biochemistry and Pharmacology, Faculty of Medicine, “Victor Babeș” University of Medicine and Pharmacy, 2nd Eftimie Murgu Square, 300041 Timisoara, Romania; cristina.marina@umft.ro (C.D.M.); vlad.daliborca@umft.ro (D.C.V.); 2Doctoral School, Faculty of Medicine, “Victor Babeș” University of Medicine and Pharmacy, 2nd Eftimie Murgu Square, 300041 Timisoara, Romania; 3Department of Cell and Molecular Biology, Faculty of Medicine, “Victor Babeș” University of Medicine and Pharmacy, 2nd Eftimie Murgu Square, 300041 Timisoara, Romania; puscasiu.daniela@umft.ro (D.P.); popescu.roxana@umft.ro (R.P.); 4Toxicology and Molecular Biology Department, “Pius Brinzeu” County Emergency Hospital, Liviu Rebreanu Blvd 156, 300723 Timisoara, Romania; cimporescu.adinela@hosptm.ro; 5Clinic of Internal Medicine-Cardiology, Klinikum Freudenstadt, 72250 Freudenstadt, Germany; moataraurica@yahoo.ro

**Keywords:** curcumin, weight loss, polyphenols, adipo-modulation, obesity

## Abstract

Obesity is not only an aesthetic problem but also an important comorbidity in metabolic syndrome and other types of pathologies. Currently discussed adjuvants are turmeric and curcumin, used as food supplements. Starting from synthesis in turmeric plant up to the use of turmeric as a spice, a significant amount of turmeric and its derivatives are lost during the processing procedure. In oral administration, the reduced bioavailability of these compounds must be taken into account, an aspect that can be improved by using different combinations and dosages. As for their pharmacodynamic effects, through its antioxidant and anti-inflammatory properties, curcumin improves mitochondrial function and promotes the browning of white adipose tissue. Another mechanism of action of curcumin in weight loss is enzymatic modulation, leading to a decrease in the activity of key enzymes involved in lipogenesis and an increase in the activity of lipolytic enzymes. These properties are enhanced by the synergistic action of the other polyphenols present in turmeric, especially calebin A, p-coumaric acid, caffeic acid and ferulic acid. Summarizing these effects, curcumin is a promising food supplement, opening new directions for further research to discover possibilities to improve or even eliminate the calamity of obesity that is currently wreaking havoc.

## 1. Introduction

Curcuma Longa, especially its product turmeric, is one of the best-known medicinal plants, frequently used in gastronomy, belonging to the antioxidant-rich ginger family (Zingiberaceae). Turmeric has been used since ancient times as an important ingredient in Ayurvedic medicine and traditional Chinese medicine [1]. The curcuma plant was brought to Europe by Arabian merchants and was even included in the travel journal of the famous explorer Marco Polo [2]. In recent decades, this substance, commonly referred to as the “golden spice”, continues to be used as a culinary ingredient, but modern technology has allowed turmeric to be applied in fields like nutrition and healthcare [3].

Even if turmeric contains a variety of bioactive constituents, current scientific interest in its therapeutic properties focuses on polyphenolic compounds, among which curcumin is the principal active compound [4]. Curcumin has been shown to have a wide spectrum of bioactive and pharmacodynamic properties including antioxidant, immunomodulatory, anti-inflammatory, antibacterial, cardioprotective, hepatoprotective, nephroprotective, neuroprotective, anti-rheumatic, anti-cancer and anti-aging effects [5]. Moreover, curcumin supplementation was found to improve obesity-related parameters, including fasting blood glucose levels and lipids, in metabolic diseases such as obesity, fatty liver and metabolic syndrome [6]. Additionally, this bioactive polyphenol compound has been explored as an adjuvant therapy for polycystic ovary syndrome because of its anti-inflammatory effects, ovulation-inducing properties and ability to improve insulin sensitivity [7]. Furthermore, curcumin has demonstrated significant potential in type 2 diabetes due to its hypoglycemic effect by delaying the absorption of carbohydrates and increasing glucose catabolism [8]. Although preclinical research has shown the therapeutic potential of Curcuma Longa, human clinical trials are crucial to ensure its good applicability in the medical field (Table 1).

The administration of curcumin is generally free of adverse effects. Over the last few years, increased attention has been given to the possibility that turmeric, frequently consumed as a food or dietary supplement, may cause adverse effects. According to recent studies, there is evidence that combining curcuma longa with black pepper, which improves the former’s bioavailability, can lead to liver injury [17,18]. In addition, there is some research in the literature indicating that turmeric can produce allergic reactions such as contact urticaria and contact dermatitis [2,19]. Moreover, excessive intake of turmeric supplements could enhance the risk of developing kidney stones, which is a major concern for susceptible patients [2]. Curcumin has been associated with a significant risk of bleeding among patients who are on warfarin and anticoagulant medications [20]. Other common adverse effects triggered by the use of turmeric supplements include nausea, diarrhea and flatulence [21,22]. Despite these aspects, most of the data from the scientific literature recognize turmeric as being well tolerated, and according to the Food and Drug Administration (FDA), it has a good safety profile [23,24,25,26].

Obesity has become a leading public health problem that affects various demographic groups, regardless of age, ethnic background and economic status [27]. Obesity represents a chronic multifactorial disease induced by excessive accumulation of adipose tissue. This condition, a major component of metabolic syndrome, has been associated with excessive reactive oxygen species (ROS) levels and elevated levels of inflammatory biomarkers, such as interleukins (IL-6, IL-18), tumor necrosis factor-alpha (TNF-α), C-reactive protein, leptin, resistin and monocyte chemoattractant protein-1 (MCP-1) [28,29]. The obesity rate has increased exponentially over the last 50 years, reaching global pandemic proportions [30]. According to the World Health Organization (WHO), it is expected that by 2025, approximately 167 million people, including both adults and children, will experience health problems caused by obesity [31]. The rapidly increasing prevalence of obesity is attracting attention due to its association with multiple comorbidities such as cardiovascular disease, renal failure, several types of cancer, Alzheimer’s disease and diabetes mellitus [32]. Equally importantly, obesity has an undeniable social and economic impact, estimated to be around USD 2.0 trillion and putting a substantial strain on healthcare systems worldwide [33]. Therefore, a comprehensive approach to obesity treatment is needed, among which the use of dietary phytochemicals (for instance, Curcuma Longa) has gained a notable attention.

In this review, we bring to the light obvious molecular and biochemical processes regarding the real involvement of biologically active phenols of turmeric in lipogenesis–lipolysis changes. This aspect is related to the transformations of these biologically active compounds, starting from their synthesis in the Curcuma plant, moving through their absorption and metabolism in the body, their actions on key enzymes and related signaling pathways, as well as the transformations that they undergo throughout this process.

## 2. From Nature to the Appearance of Biologically Active Compounds: Biosynthesis of Curcumin and Other Related Phenols in Curcuma

To understand a molecule’s structure–effect relationship, especially when it comes to natural compounds, one must look at its overall and natural synthesis phase. This is important because, unlike synthetic products where the exact amount of substance(s) administered is known, in products extracted from biological matrices, the production is influenced by environmental conditions, soil, and precipitation, but also by other species in the vicinity.

The biosynthesis of curcumin and its derivative compounds is carried out via the polyketide pathway, but also via the phenyl-propanoid pathway, without being able to make an exact delimitation, as founded in other plant species such as Cannabis sativa, where, each synthesis pathway yields distinct compounds [34]. The synthesis starts from phenylalanine with the transformation to p-coumaric acid passing through the intermediate step of cinnamic acid. The two steps are carried out by phenylalanine-ammonia lyase and cinnamate-4-hydroxylase. p-coumaric acid can be activated directly to p-coumaryl-CoA or it can go through intermediate steps to caffeic acid, ferulic acid and then activation to feruloyl-CoA. As a side reaction, caffeic acid can be activated to caffeoyl-CoA and then converted to feruloyl-CoA. All activations are catalyzed by 4-coumarate-CoA ligase [35,36,37] (Figure 1a). In the second step, after the precursors formation, they will condense with malonyl-CoA resulting in two diketide compounds, p-coumaroyl-CoA-diketide and feruloyl-CoA-diketide. In the next step, both precursors condense with two diketides obtaining enol forms of Bis-demethoxycurcumin, Demethoxycurcumin, known as curcuminoids, and Curcumin. Following tautomerization, enol forms of Curcumin and derivatives will be transformed into keto forms [36,37,38,39] (Figure 1b).

Trace amounts of curcumin will be transformed into Calebin A in turmeric rhizomes [40]. Calebin A, a stable compound, is actually a ferulate ester obtained from a Baeyer–Villiger reaction, and not from an esterification reaction. Baeyer–Villiger-type monooxygenases catalyze insertion of oxygen into a C-C chain in the vicinity of keto groups. This reaction is carried out in the presence of microorganisms such as *Ovatospora brasiliensis*, which possess this enzyme [41] (Figure 2). Although it is found in small quantities, some studies attribute antiproliferative, antioxidant and anti-inflammatory properties to it [42,43] probably due to its polyphenol-type structure and also in the context of the global potentiation of all polyphenols, where the main effects are attributed to overall effect of polyphenols (e.g., honey) [44,45,46]. Thus, all intermediate products carrying the phenolic OH group may contribute to the antioxidant properties of turmeric and turmeric extract.

## 3. From Exogenous Intake to Pharmacokinetics: Pharmacokinetics of Curcumin and Phenolic Derivatives in the Body

A lot is known about turmeric, curcumin, and curcuminoids, but an important aspect would be the amount of the Curcuma plant compounds that actually reaches the tissues.

In most cases, the product intended for consumption, whether it is used as a spice or as a food supplement, uses turmeric. Turmeric is the powder obtained from curcuma rhizomes after the dehydration and drying process (traditionally under sunlight). Dehydration, in general, is a method of preservation and prevention of spoilage [47,48]. Moreover, with drying, the yellow-orange color also intensifies due to curcuminoid pigments. A study showed that sun drying degrades 72% of curcumin and curcuminoids, mainly due to the effect of light and UV radiation, while freeze drying and hot-air drying show a degradation of 55% and 61%, respectively [47]. When covered with a polycarbonate cover at 40 °C, it was observed that it has a protective effect against light and UV radiation, with the lowest degradation of curcumin and curcuminoids [49].

The chemical structure of curcumin and its derivatives results in low water solubility and reduced digestive tract absorption. They are also unstable in aqueous solution and degrade rapidly at physiological pH by autoxidation to bicyclopentadine, vinylether and siroepoxide [50]. At alkaline pH, they decompose to vanillin, ferulic acid, ferulic aldehyde, feruloyl-methane, and trans-6-(4′-hydroxy-3′-methoxyphenyl)-2,4 dioxo-5-hexanal [50,51]. Bioavailability is also reduced by the first hepatic passage. The half-life has been estimated at 10 min. Furthermore, the hepatic metabolism of glucuronide and sulfoconjugates is quick, and they are quickly removed by biliary and renal systems without accumulating in any organ [51,52]. Despite the fact that some authors do not consider bioavailability to be an important factor for therapeutic efficacy [53], there are already a variety of ways to increase bioavailability. Some studies focus on improving bioavailability when turmeric is used as a spice, while others concentrate on the bioavailability offered by pharmaceutical formulations when it is used as a food supplement.

The therapeutic effects of curcumin are promising, but there is an essential factor that must be considered: it has limited bioavailability due to its lipophilic molecule, fast metabolism and rapid systemic clearance [15,54]. In recent years, scientists have tried to find ways to improve the bioavailability of curcumin in order to benefit from its medical properties. One of the most used combinations is the association with piperine (a compound from black pepper), which increase the absorption of curcumin and greatly improve its bioavailability, but requires strict monitoring because it also increases the absorption of other drugs and reduces hepatic metabolism as well [2,15].

Curcumin nanomedicine formulations, including nanoemulsions, solid lipid nanoparticles, nanocomposite, liposomes, micelles and mixed micelles, hydrogels, polymeric nanoparticles and other, have been intensively investigated [55]. Clinical research has demonstrated that the application of nanotechnology is extremely effective as a drug delivery system, eliminating natural obstacles, ensuring better absorption and distribution of curcumin, increasing the circulation time, improving permeability and its resistance to the metabolic degradation [56]. Another efficient strategy to enhance the water solubility of curcumin is the creation of a supramolecular cyclodextrin-curcumin complex [57]. Furthermore, the association with lycopene amplifies the antioxidant capacity, and also co-administration with glycyrrhetinic acid could have antiproliferative effects on liver cancer cells and induce apoptosis [2]. Among these strategies, we can mention the use of metal ions such as Zn^2+^, Cu^2+^, Mg^2+^, and Se^2+^, which are able to form complexes with curcumin in order to improve its pharmacokinetic properties [2].

Thus, it can be seen that, after the ingestion of natural curcumins such as those found in turmeric, the amount available for pharmacological effects is extremely small. In addition to losses during the processing of turmeric to spice form, there is reduced absorption and low persistence in the body. Improving the natural bioavailability of curcumins by combining them with other spices does not bring a significant contribution. In order to achieve a desirable effect, administration in the form of pharmaceutical formulations is the only option.

## 4. What Remains to Be Done from Pharmacokinetics to Pharmacodynamics: Studies That Explain the Adipo-Modulation Action of Curcumin and Phenolic Derivatives

The anti-obesity effect has been studied over time, revealing various mechanisms that could be explained. However, the net efficacy of curcumin and its derivatives in the treatment of obesity has been questioned in some studies [58,59]. However, there are numerous hypotheses that confirm the anti-adipose effect of curcumin: anti-inflammatory and antioxidant effect, enzymatic modulator in lipid metabolism, interference with certain cellular signaling pathways in order to accumulate adipose tissue and, last but not least, stimulation of consumption by influencing thermogenesis.

### 4.1. Modulation of Adipogenesis Through Antioxidant and Anti-Inflammatory Capacity

It is known that the accumulation of ROS reagents causes alterations of the structure of lipids, proteins, and DNA, but also of cellular organisms, especially mitochondria, which are the source of ROS. In fact, it is considered that the major source of ROS production is mitochondria and enzymes of the type NADP-oxidase that transform NADH+ H^+^ and NADPH+ H^+^ in NAD^+^ and NADP^+^, respectively [60], which will then be converted to reduced forms through nicotinamide nucleotide transhydrogenase (NNT) in mitochondria [61]. In obese people, mitochondrial dysfunction is associated with adipogenesis, reducing the oxidation of free fatty acids and activating biosynthesis of fatty acids with lipid accumulation [29]. An excess of NADP-oxidase becomes dysfunctional because it leads to excessive formation of ROS initiating intense lipid peroxidation in obese individuals [62]. On the other hand, obese individuals show a low antioxidant capacity in the sense of reduced activity of superoxide dismutase (SOD), glutathione peroxidase (GPX), catalase and NNT [63]. It has been observed that curcumin can scavenge superoxide radicals, NO and H_2_O_2_ preventing lipid peroxidation due to increased activity of SOD and catalase [64]. In a study conducted on male albino rats fed with a high-fat diet, it was observed that the addition of curcumin to the diet caused an increase in the activity of SOD, GPX, catalase and the amount of reduced glutathione (GSH), as well as a decrease in hepatic myeloperoxidase (MPO), considered a risk marker for obesity [65]. Furthermore, it was demonstrated that curcumin is able to improve mitochondrial respiratory function by upregulating peroxisome-activating receptor γ (PPAR-γ), a nuclear receptor involved in adipogenic differentiation [66]. In this way, it seems that the improvement of mitochondrial function would play an essential role in the conversion of white adipose tissue (WAT) into beige adipose tissue through browning or even into brown adipose tissue (BAT) where, also at the mitochondrial level, uncoupling protein 1 (UCP-1) is stimulated by curcumin, especially in BAT [67,68,69].

Upregulation of PPAR-γ by curcumin in the context of inflammation and ROS generation causes a heterodimerization with retinoid X receptor (RXR), the molecular assembly formed will bind to peroxisome proliferation response element (PPRE) followed by induction of gene transcription and suppression of proinflammatory cytokine release [70,71]. Like any protein structure, PPAR-γ undergoes posttranslational modifications of which the most important is acetylation. PPAR-γ acetylation in BAT causes mitochondrial dysfunction, decreased UCP-1 expression and whitening. Intense acetylation is associated with obesity and aging [72]. In the browning phenomenon, PPAR-γ deacetylation by sirtuin-1 (SIRT-1) is observed [73], curcumin stimulating SIRT-1 thus promoting PPAR-γ deacetylation [72,74]. Furthermore, SIRT-1 deacetylates and activates peroxisome proliferator-activated receptor G coactivator 1-α (PGC-1α) and promotes mitochondrial biogenesis [71], and also reduces the translocation of dynamin-related protein 1 (DRP-1) from cytoplasm to the mitochondria, reducing mitochondrial fragmentation and preserving mitochondrial integrity [75] (Figure 3).

In addition, where there is oxidative stress, there is also inflammation because these two events are associated—ROS produces inflammation and inflammation releases ROS. Inflammation produced by ROS activates the production of nuclear factor-kB (NF-kB) and adipokines such as IL-13, IL-10, IL-4, transforming growth factor β (TGF-β) and also proinflammatory factors such as TNF-α and IL-6 [58,76]. Curcumin’s inhibition of NF-kB results in the depression of these mediators responsible for initiating and maintaining the inflammatory process [77] (Figure 4), an effect also demonstrated in THP-1 cell cultures and foam cell models [78]. In 3T3-L1 adipocytes stimulated with the bacterial endotoxin lipopolysaccharide to induce inflammatory phenomena, curcumin supplementation reduced IL-6 expression, the authors suggesting that the effect would be produced by upregulation of the mammalian target of rapamycin (mTOR) pathway [79]. In addition, curcumin suppresses macrophage inhibitory protein and MCP-1, the major factor for macrophage chemotaxis [80]. Thus, curcumin plays a major role in decreasing WAT infiltration by macrophages, reducing the main factor involved in the release of adipokines [81] and initiation of BAT apoptosis [82].

Microbial infections are also accompanied by inflammatory phenomena. A number of studies demonstrate the antimicrobial effect of turmeric and curcumins. Among microbial species, an antibacterial effect has been observed on some Gram-positive strains, for example, *Staphylococcus aureus* [83,84,85], *Enterococcus fecalis* [85], Gram-negative ones such as *E. coli* [83,85], *Helicobacter pylori* [83], *Proteus mirabilis*, *Pseudomonas aeruginosa* [85] acid-fast, and *Mycobacterium tuberculosis* [83], and also on Candida species [83,86]. Furthermore, there are studies that have investigated the relationship between curcumin and gut microbiota (GM). Curcumin is degraded in the digestive tract by local bacterial flora into tetrahydrocurcumin, dihydroferulic acid, dihydrocurcumin, and hexahydrocurcumin [1], but also exerts a protective effect on these microorganisms, e.g., Bacteroidaceae, Rikenellaceae [87,88], Bifidobacterium, and Lactobacillus [88]. Thus, curcumin and curcuminoids could indirectly promote weight loss, since GM can regulate monosaccharide absorption from the digestive tract and modulate hepatic lipogenesis [88,89]. It has been shown that HFD depresses beneficial GM and stimulates hepatic steatosis [89]. In a study conducted on B6.V-Lep ob/obJRj mice treated with 0.3% curcumin, it was demonstrated that turmeric promotes the multiplication of *Bifidobacterium* spp., *Lactobacillus* and contributes to alleviation in hypertriglyceridemia after 5 weeks [90]. In addition, among metabolites, tetrahydrocurcumin has an important therapeutic role by improving hepatic steatosis in obese C57/BL/6 mice [91].

In this regard, turmeric may reduce oxidative stress and inflammation through the modulation of local pathways that increases the ability of adipose tissue to remove free radicals but also has a noteworthy potential to promote WAT browning. Moreover, through its protective effect on GM, it makes an additional contribution to the improvement of hyperlipidemia, obesity and metabolic syndrome. Starting from this idea, a perspective can be opened to explore the synthesis of pharmaceutical compounds that would be capable to achieve this transformation in people with a tendency towards overweight to accomplish the prophylaxis of obesity and associated consequences.

### 4.2. Effect on Lipid Metabolism

The accumulation of adipose tissue along with the damage caused by ROS, especially at the mitochondrial level, favors lipogenesis and reduction in β-oxidation. It has been demonstrated that administration of curcumin as a dietary supplement can decrease the plasma concentration of LDL cholesterol and TGL with the reduction in abdominal fat. The observed results are attributed to decrease in fatty acid synthase (FAS) activity and increase in expression of PPAR-α, and carnitine palmitoyltransferase-1 (CPT-1) [64]. The increase in CPT-1 activity will improve the entry of fatty acids into mitochondria while PPAR-α will promote β-oxidation of fatty acids and cholesterol catabolism [92]. In fact, acetyl-CoA is the precursor molecule of both cholesterol and fatty acids that will later be esterified to TGL. Lipid synthesis is also modulated by 5′-AMP-activated protein kinase (AMPK) which reduces fatty acid synthesis by interfering with the activity of FAS, sterol regulating element binding proteins (SREBPs) and 3-hydroxy-3-methylglutaryl-CoA reductase, suppressing the production of cholesterol and fatty acids [58,93,94,95,96]. It is currently considered that AMPK is the control center of lipid metabolism that determines a decrease in acetyl-CoA carboxylase (ACC) activity through its phosphorylation, and reduces conversion of acetyl-CoA to malonyl-CoA, while increase CPT-1 activity [93,94,95,96,97]. When curcumin was administered to Wistar rats, its renal protective effect was observed regarding the development of dyslipidemia through the activation effect on PPAR-α and CPT-1, restoring β-oxidation and also reduces hepatic and renal lipid synthesis [98]. In this regard, following curcumin administration, several studies have reported decreases in plasma total cholesterol, TGL, LDL-cholesterol, VLDL-cholesterol [99,100,101] along with cholesterol efflux from adipocytes [102]. These beneficial effects on the lipid profile were observed at doses of 1200 mg/day, administered for 12 months [101,103].

Changes in plasma concentrations of TGL, total cholesterol and cholesterol fractions also have a positive effect on slowing down/improving atherosclerosis process by reducing the formation of oxidized LDL and also by reducing the formation of foam cells [104,105].

Therefore, doses of 1200 mg/day of curcumin or higher, administered as a food supplement, in addition to enhance lipolysis, reduce lipogenesis and body mass, also intervene in optimizing lipid profile, with a significant potential in improving atherosclerotic phenomena.

### 4.3. Interferences During Adipogenesis

The plasticity, or more correctly, elasticity (quickly returns to its previous state as soon as the modifying factor has ceased) of adipose tissue makes it responsive to numerous influences, changes being reversible under the action of external factors or medications.

Adult adipose tissue is divided into WAT and BAT. BAT is involved in thermogenesis, containing a large number of mitochondria, while WAT is the main form of energy storage and has few mitochondria [106]. WAT can be visceral or subcutaneous: in the visceral type, the expansion is mainly by hyperplasia, while in the subcutaneous type, hypertrophy predominates [107]. An intermediate category is represented by beige adipose tissue (BeAT) with a higher number of mitochondria and a higher thermogenic activity compared to WAT [107,108]. The transformation of WAT into BAT, initially acquiring the appearance of BeAT, a process called browning, contributes to weight loss and improvement of overweight and obesity. The browning process is accompanied by an increase in the number of multilocular lipid droplets and an increase in UCP-1 expression [109,110].

Adipogenesis is generally controlled by PPARs, CCAAT/enhancer-binding proteins (C/EBPs), SREBPs, and by lipogenic enzymes ACC and FAS [111,112]. Among these factors, the main regulator of adipocyte differentiation is PPAR-γ, which occurs in two forms: PPAR-γ1 expressed in all tissues except muscle and PPAR-γ2 expressed only in adipose tissue and the intestine [71,113]. The most adipogenic form is PPAR-γ, the form induced by HFD: weight gain appears to be caused by PPAR-γ under conditions of excess energy [73]. The activation of PPAR-γ in adipocytes determines lipogenesis with increased expression of C/EBP-α [114] as well as metabolic phenomena such as the release of free fatty acids (FFA) from lipoproteins, intracellular transport of FFA, esterification with glycerol and formation of TGL [114,115,116]. Reductions in PPAR-γ and C/EBP-α expressions by curcumin contribute to the reduction in TGL accumulation in adipocytes and a decrease in plasma concentrations of cholesterol and TGL [112] (Figure 5). Turmeric extract applied to 3T3-L1a cells showed a reduction in the activity of ACC, FAS enzymes as well as downregulation of PPAR-γ and C/EBP-α expressions [111]. Another study on the same type of cells demonstrates the triggering of preadipocyte apoptosis produced by curcumin at high doses of over 30 µM on both extrinsic and intrinsic pathways by activating caspases 8, 3, 9 while at low doses of 15 µM, the expression of PPAR-γ and C/EBP-α was inhibited. The effect was evident after incubation for 24 h, being maximal at 72 h and 48 h, respectively [117].

It is known that adipose tissue exhibits a rebound phenomenon in weight gain after various restrictive diets [118,119,120], as well as a significant capacity for its expansion and proliferation [121,122]. Transition from in vitro experiments to in vivo use in humans would involve chronic administration of turmeric/curcumin in doses that are constantly requires a sustaining high tissue concentrations. Furthermore, continuous administration over long periods of time is necessary to maintain the achieved results, taking into account that adipose tissue exhibits a rebound of weight gain when the inhibitory factor has been removed.

### 4.4. The Role of Curcumins in Increasing Energy Utilization and Stimulating Thermogenesis

Thermogenesis is actually the main phenomenon by which weight loss is achieved due to increased energy expenditure. Thermogenic adipose tissue is represented by BAT and BeAT, which have a large number of mitochondria and an increased expression of UCP-1 that uncouples ATP production and generates heat [123,124]. Another agent that induces browning is PGC-1α, and the absence of PGC-1α expression stops the transition from WAT to BeAT [125]. Upregulation of PGC-1α defines activation of PPARs [126]. It has been observed that administration of curcumin at a dose of 100 mg/kg body weight to different types of mice that received a high-fat diet (HFD) improves the browning process of WAT, and thermogenesis reduces the size of adipocytes and increases the expression of PGC-1α, PPAR-γ and UCP-1 [127,128]. Another study conducted on Sprague Dawley rats where curcumin was added as a supplement to the daily diet revealed a lower body weight, less adipose tissue and increased energy expenditure. Increased expression of UCP-1 was observed in adipocytes in these rats, and compared to the group without curcumin supplementation, it was observed that UCP-1 (being a protein associated with an increased thermogenesis) also represents an indicator of WAT browning and BAT activation [129].

Browning WAT and activation of thermogenesis are important phenomena triggered by curcumin, especially in patients who cannot improve their physical activity and as an adjunct in weight loss diets. Although there are studies on laboratory animals, the importance of the impact of these phenomena produced by curcumin on humans leaves room for further research and improvements.

## 5. Minor Amounts–Major Importance: The Role of Calebin A and Intermediates in the Global Effect of Turmeric

Minor components and intermediate compounds can be found in turmeric and may contribute to the overall pharmacological effect. Among these, importance is given to Calebin A as well as the intermediates coumaric acid, caffeic acid and ferulic acid.

Calebin A, a non-toxic compound, can suppress adipogenesis by downregulating PPAR-γ, and C/EBP in adipocytes with inhibition of adipocyte differentiation. Calebin A is also able to induce lipolysis and inhibit lipogenesis by decreasing FAS activity [130]. Regarding its action at the intestinal level, it behaves as a dipeptidyl peptidase IV inhibitor with an effect on the incretin system, running as an effective antidiabetic in type 2 diabetes mellitus but also in the treatment of obesity [131]. C57BL6/6J mice fed with a HFD supplemented with Calebin A showed a reduction in glycemia, adipose tissue and an enhancement of thermogenesis [132]. As an anti-inflammatory effect, Calebin A suppresses production of pro-inflammatory cytokines TNF-α, NF-kB [133] as well as decreases the enzymatic activity of cyclooxygenase and lipoxygenase [134].

p-Coumaric acid, a phenol identified in Curcuma longa rhizome [135] and the major secondary metabolite in leaves and stems of Curcuma longa, found in an amount of 0.431 mg/g [136], is an intermediate compound that contributes to the anti-obesity effect. In C57BL/6J mice fed with a HFD to which p-coumaric acid was added at a dose of 10 mg/kg body weight/day for 16 days, a reduction in WAT and an increase in fatty acid β-oxidation were observed, as well as an improvement in the general lipid profile [137]. Unlike curcumin and Calebin A, p-coumaric acid, caffeic acid and ferulic acid are readily absorbed in the rat intestine via a monocarboxylic acid transporter [138]. If Curcuma longa is fermented, a large amount of caffeic acid is found along with curcuminoids. The fermentation mixture product showed an inhibitory effect on SREBP and ACC as well as a reduction in PPAR-α and CPT-1 activity [139]. In addition, caffeic acid can diminish intracellular lipid accumulation [140]. Ferulic acid can reduce adipocyte size, body weight, and visceral fat accumulation [141]. Ferulic acid supplementation in HFD-fed mice attenuated hypercholesterolemia by increasing cholesterol-7α-hydroxylase expression and enhancing digestive bile acid clearance [142]. Ferulic acid administration at 5 g/kg for 8 weeks reduced body weight in these mice and improved lipid profiles [143]. Similar results were observed at 2 g/kg of ferulic acid supplementation in mice fed with the same diet, with anti-obesity effects alongside antioxidant and anti-inflammatory properties [144].

In addition to curcumin, curcuminoids and intermediates shown in Figure 1 and Figure 2 (p-coumarinic acid, caffeic acid, ferulic acid, calebin), there are a number of products considered minor but with an important contribution to the overall bioactivity of turmeric, in general [145,146,147,148]. These components are summarized in Table 2.

It seems that in natural products, the pharmacodynamic effect cannot be attributed to just one compound, even if it is predominant. It can be observed that the effects of the other components are similar, probably the overall pharmacodynamic response is the result of the synergistic action between the major and minor components.

## 6. Discussions and Perspectives on the Real Weight Loss Effect

Although in vitro and in vivo studies on laboratory animals show an effect of reducing adiposity and weight loss, curcumin raises questions about its effectiveness in humans.

In a systematic review where a meta-analysis of the literature data was performed and 193 studies were evaluated, of which only 2 remained eligible, it was found that curcumin can be an adjuvant in weight loss only if it is administered in pharmaceutical forms that improve bioavailability [165]. In another review, 1163 articles were analyzed, of which only 18 articles relevant to the authors remained, which included a total of 1604 people. Here, curcumin produced a decrease in body mass index, weight, waist circumference, leptin and an increase in adiponectin was noted, but without any impact on hip circumference [166]. Anthropometric indices showed changes only if 1500 mg curcumin/day was administered but not visible if doses of 1000 mg/day or less were used. In contrast, phytosomal curcumin form used in doses of 200–400 mg/day for 2–4 months showed significant reductions in anthropometric indices [167]. Administration of 1 g turmeric + 1 g black pepper/day together with each breakfast, reduced postprandial blood glucose values and increased the feeling of satiety in the 10 men and 10 women studied [168]. Combination of 500 mg curcumin + 30 mg zinc/day for 90 days administered to 20 patients as tablets, in a dosage form with increased bioavailability, produced an improvement in lipid profile in terms of decreasing TGL, LDL, and increased HDL level, but without influencing total cholesterol values, feeling of hunger, waist and hip circumferences [169].

Despite a large number of research and review articles on the use of dietary supplements in general, their administration is not accompanied by increased efficacy, persistent effects over time and clear results demonstrating significant weight loss. Batsis J.A. et al. [170] underline this in a systematic review, even though, it is well known, dietary supplements, especially antioxidants, have their well-defined role as adjuvants being included in some therapeutic strategies even in some types of cancer [171,172,173,174]. Other limitations of the literature study that addresses research on human subjects consist in the transposition of theoretical knowledge and in vivo experiments on laboratory animals to human applications, where the phenomena observed in controlled laboratory experimental conditions are irrelevant in the population studied. In general, studies conducted on human subjects were carried out for limited periods of time under different types of standardized diets throughout the research period in order to have a reference comparison. In real life, these diets are not adopted for a long time, appetite being variable depending on the season, different physiological or pathological states as well as people financial situation, which nowadays primarily determines the quality of the food consumed. Even if high doses demonstrate significant therapeutic efficacy and, in some cases, modest efficacy, curcumin should not be considered a “magic pill” that causes major weight loss or prevents weight gain in case of an increased and unbalanced food intake. Also, after stopping the administration of curcumin preparations, the gains obtained are probably lost in a variable period of time, taking into account the well-known rebound effect of weight loss.

These promising results obtained by using curcumin in weight loss therapy represent only the beginning of research in the field. Based on the findings presented above, as well as the partial elucidation of some molecular mechanisms, the possibility of further exploration of multiple variants and combinations that would fulfill the desire of most people to be and to stay young, attractive and slim can be opened.

## 7. Conclusions

Obesity is a major problem nowadays that causes a multitude of challenges and sometimes therapeutic failures. Among the natural alternatives is the use of turmeric or curcumin as a food supplement. Turmeric polyphenols, especially curcumin and its derivatives, have shown to be candidates to be taken into account in terms of their use as adjuvants in anti-obesity treatments. On the one hand, they mediate mitochondrial protection, the place of cellular respiration and β-oxidation of fatty acids, due to their antioxidant and anti-inflammatory properties. Improving this aspect, curcumin can trigger the browning process of WAT. Along with the activation of BAT and BeAT, the thermogenesis process is improved, and displays an important role in weight loss. On the other hand, curcumin has the ability to modify the activity of some enzymes involved in the lipogenesis-lipolysis process with a shift of the balance towards lipolysis. Although at first glance the effects on PPAR-γ are contradictory, as evidenced by the research presented, the action of curcumin is context-dependent, the complete intimate mechanism is not yet being elucidated. Nevertheless, the final product will always lead to improved mitochondrial function, lipolysis and promotion of weight loss. In humans, studies conducted have not accurately reflected the results observed in vitro or using lab animals. This is probably due to the complexity of the enzyme systems and cellular regulatory and signaling pathways, and also to the limited time of the research. Consequently, it will be necessary to identify conditions and formulations that will allow laboratory to find greater similarities in human subjects. However, the results are promising, even if sometimes controversial, but they generate challenges for further research.

## Figures and Tables

**Figure 1 ijms-26-06880-f001:**
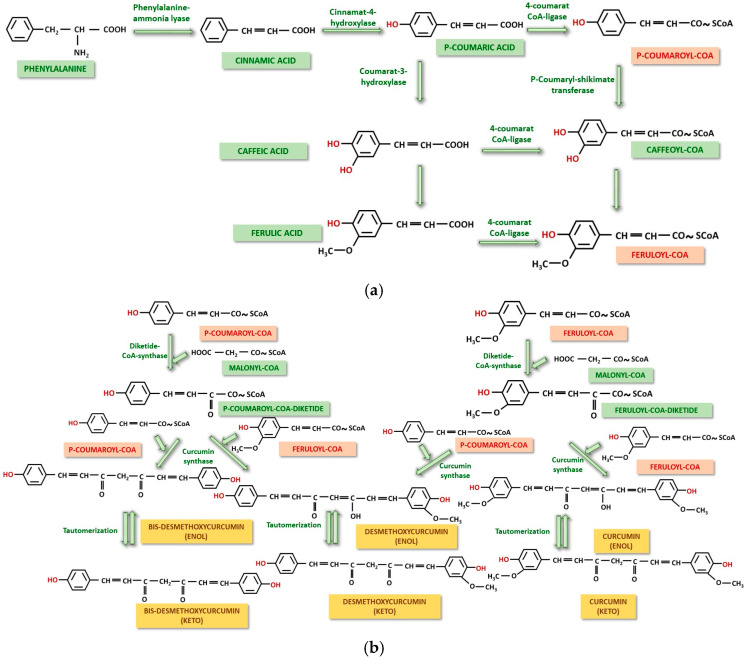
Biosynthesis of curcumin and its derivatives starting from phenylalanine. Synthesis of the precursors p-coumaryl-CoA and feruloyl-CoA, marked in red. Phenolic OH groups are also marked in red (**a**); interconversions of precursors to enol forms of Bisdesmethoxycurcumin, Desmethoxycurcumin and Curcumin, and enol-ketotautomerization are schematically presented. Phenolic OH groups and precursors p-coumaryl-CoA, and feruloyl-CoA are marked in red (**b**).

**Figure 2 ijms-26-06880-f002:**
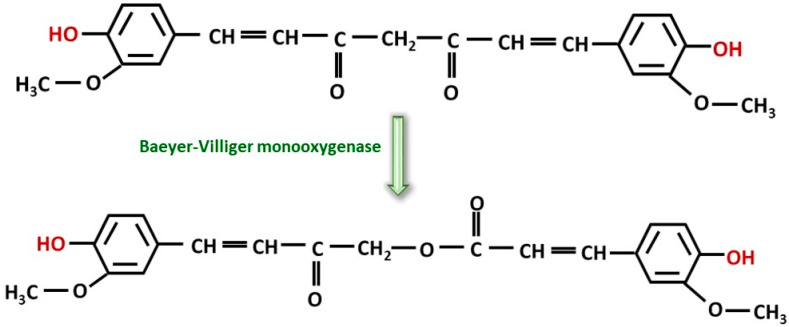
The transformation of curcumin into calebin A is carried out by catalysis of Baeyer–Villiger-type monooxygenase.

**Figure 3 ijms-26-06880-f003:**
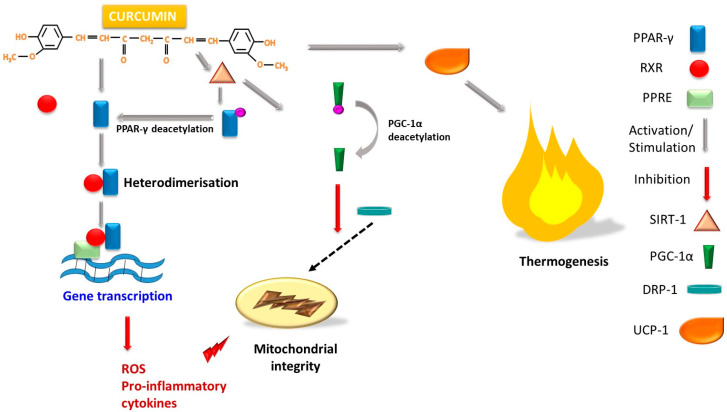
Schematic representation of the effect of curcumin on upregulation of PPAR-γ and promotion of thermogenesis through activation of UCP-1. PPAR-γ—peroxisome-activating receptor γ; UCP-1—uncoupling protein-1; DRP-1—dynamin-related protein 1; SIRT-1—sirtuin 1; RXR—retinoid X receptor; PPRE—proliferation response element; PGC-1α—peroxisome proliferator-activated receptor G coactivator 1-α.

**Figure 4 ijms-26-06880-f004:**
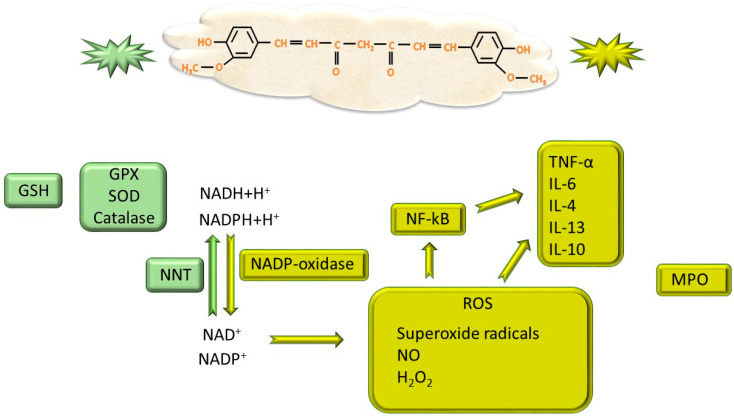
The changes produced by curcumin. Green illustrates the activating effect on enzymes involved in the removal of free radicals; yellow shows the depressing effect that curcumin exerts on ROS, ROS-producing factors, as well as pro-inflammatory mediators. ROS—reactive oxygen species; MPO—myeloperoxidase; SOD—superoxide dismutase; GPX—glutathione peroxidase; NNT—nicotinamide nucleotide transhydrogenase; GSH—reduced glutathione; NF-kB—nuclear factor-kB.

**Figure 5 ijms-26-06880-f005:**
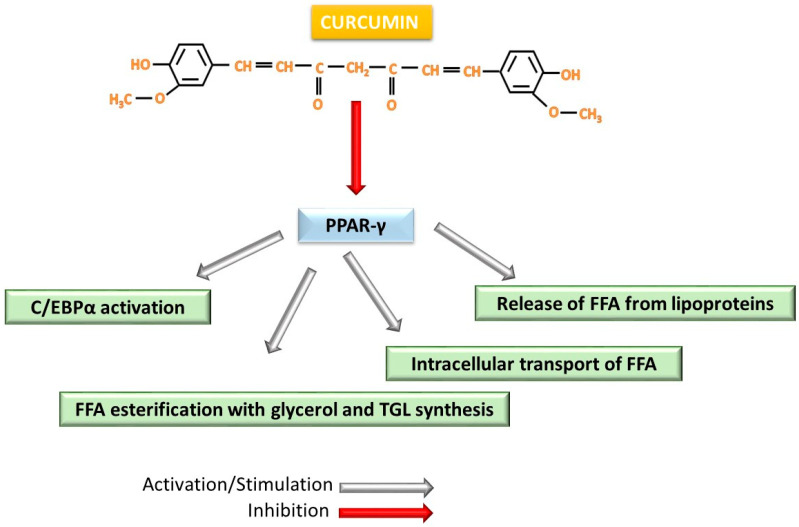
Schematic representation of the inhibitory action of curcumin on PPAR-γ and the main biochemical processes stimulated by PPAR-γ in adipogenesis. PPAR-γ—peroxisome-activating receptor γ; C/EBPα—CCAAT/enhancer-binding protein α.

**Table 1 ijms-26-06880-t001:** Human trials involving curcumin and their main outcomes.

Population	Study Type	Intervention	Outcome	Duration	References
Middle-aged and elderly overweight participants	Randomized, double-blind, placebo-controlled	Turmeric extract mixture (CLE)	- Body weight, BMI and inflammatory markers decreased - Mental health score and anger–hostility improved	12 Weeks	[9]
Subjects with mild/moderate knee joint pain	Pilot, randomized, double-blind, placebo-controlled clinical trial	B-Turmactive^®^ (a new formulation of dry extracts of turmeric roots) and brewer’s yeast as a placebo	- Reduced global pain; however, only B-Turmactive^®^ had superior effects in reducing night-time pain	1 week	[10]
Obese subjects with type-2 diabetes	Randomized, double-blind, placebo-controlled trial	Curcumin extract 1500 mg/day	- Important decrease in fasting blood glucose and HbA1c - Improved function of β-cells - BMI was lowered	12 months	[11]
Healthy males 20–40 years old	Randomized, double-blind, crossover pilot trial	Curcuma longa powder; low dose (1.5 g), moderate dose (3 g), high dose (6 g), separated by 7-day washout	- High doses of C. longa did not substantially influence the antioxidant effect. Moreover, lower doses might optimize antioxidant capacity	3 weeks	[12]
Mild to moderate hospitalized COVID-19 patients	Triple-blind randomized, placebo-controlled	Oral nanocurcumin formulation (Sinacurmin^®^ soft-gel 40 mg)	With the exception of a sore throat, symptoms were reduced considerably faster compared to the placebo treatment	4 months	[13]
Female subjects with polycystic ovary syndrome	Randomized, double-blind, placebo-controlled trial	Curcumin 500 mg three times daily	Significantly decreased fasting plasma glucose and insulin, reducing sex hormone levels and hirsutism score	12 weeks	[14]
Patients with knee osteoarthritis (OA)	Randomized, non-inferiority, controlled clinical trial	Bioavailable turmeric extract (BCM-95^®^) 500 mg capsules versus paracetamol 650 mg tablets	The results obtained considering bioavailable turmeric extract were comparable to those observed with paracetamol, improving physical function and pain relief in patients with knee OA	6 weeks	[15]
Patients with nonalcoholic fatty liver disease	Randomized, double-blind, placebo-controlled clinical trial	Turmeric 2 g/day	- Substantial reduction in liver enzymes (AST, ALT, GGT) - Significant decrease in triglyceride and LDL, HDL, MDA levels - No significant change in liver echogenicity, total cholesterol and VLDL levels	8 weeks	[16]

**Table 2 ijms-26-06880-t002:** Other components found in small amounts in turmeric and their main biological activities.

Component	Structure	Main Activity
*Turmerones*: Aromatic (Ar)-turmerone	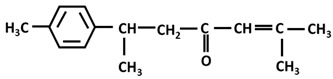	- arrest proliferation and decrease mobility of glioma cells [149] - prevention dopaminergic neurodegeneration induced by microglial activation [150]
α-turmerone	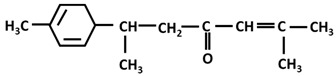	- anti-inflammatory [151,152]
β-turmerone	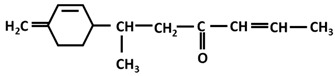	- anti-inflammatory [152]
*Sesquiterpenes*: α-curcumene	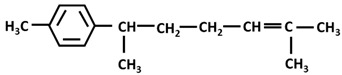	- antibacterial (*Staphylococcus aureus* and *E. coli*) [153] - anti-proliferative (lung adenocarcinoma cells) [154]
Curcumanolide	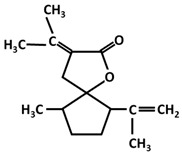	- anti-inflammatory [151]
Zingiberene	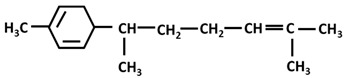	- antibacterial (*Staphylococcus aureus* and *E. coli*) [153] - anti-proliferative (liver cancer cells) [155]
*Germacrane*: Germacrone	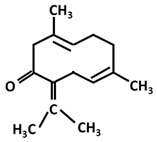	- inhibition of collagen-induced platelet aggregation [156] - anti-proliferative effect on prostate cancer cells [157] and esophageal squamous cell carcinoma cells [158]
Germacrene	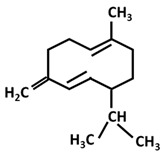	- anti-inflammatory [159] - cytotoxic activity against liver and breast carcinoma cells [159]
*Phenols/polyphenols*: Quercetin	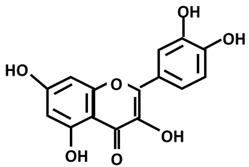	- reduces viral replication in synovial fibroblast cells infected with O’nyong-nyong virus [160] - adjuvant in hepatic fibrosis [161]
Vanillin	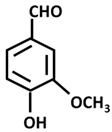	- spasmolytic effect [162]
*Guaiane*: Curcumenol	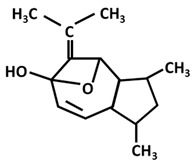	- anti-tumoral (lung cancer, liver cancer, gastric cancer [163], breast cancer [164]), hepatoprotective [163]
Procurcumenol	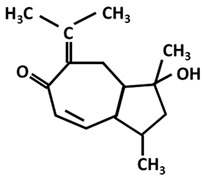	- inhibition of collagen-induced platelet aggregation [156]

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
