# Peer review of "Adipo-Modulation by Turmeric Bioactive Phenolic Components: From Curcuma Plant to Effects"

_ijms, 2025, doi:10.3390/ijms26146880_

Round 1
Reviewer 1 Report
Comments and Suggestions for Authors
This manuscript presents a comprehensive review of the potential anti-obesity effects of turmeric-derived phenolic compounds, with a focus on curcumin and its related metabolites. The authors explore the biosynthesis, pharmacokinetics, and pharmacodynamic actions of these compounds, highlighting mechanisms related to adipose tissue modulation, inflammation, thermogenesis, and lipid metabolism. The manuscript is interesting and well written and is of high scientific interest. However, some issues regarding language, clarity, and clinical context need to be addressed:
1. The discussion of human trials is relatively brief and lacks critical engagement. Please expand the section on clinical evidence to assess sample sizes, endpoints, intervention design, and statistical robustness of the included studies. A summary table of key clinical trials would also be helpful.
2. Some mechanistic explanations (eg. the role of antioxidant enzymes or NF-κB inhibition) appear multiple times in different sections. Please reduce redundancy by integrating overlapping discussions in Sections 4.1 (oxidative stress) and 4.2 (lipid metabolism).
3. The role of PPAR-γ is described inconsistently (as both upregulated and downregulated) depending on context. While such context-dependence is plausible, it requires clearer framing to avoid confusion. A summary figure or table clarifying which effects occur under which conditions (e.g., dose, model system, tissue type) would improve clarity.
4. Since there are limited human clinical data, please moderate/adjust the conclusion to better reflect the current evidence, emphasizing that although preclinical results are encouraging, larger and longer-term clinical trials are needed to substantiate efficacy in humans.
5. A thorough linguistic revision/editing is necessary for academic precision.
Comments on the Quality of English LanguageA thorough language/linguistic revision/editing is necessary for academic precision.
Author Response
Comment 1: The discussion of human trials is relatively brief and lacks critical engagement. Please expand the section on clinical evidence to assess sample sizes, endpoints, intervention design, and statistical robustness of the included studies. A summary table of key clinical trials would also be helpful.
Response: We introduced Table 1 containing clinical trials on humans, as requested by the referee.
Comment 2: Some mechanistic explanations (eg. the role of antioxidant enzymes or NF-κB inhibition) appear multiple times in different sections. Please reduce redundancy by integrating overlapping discussions in Sections 4.1 (oxidative stress) and 4.2 (lipid metabolism).
Response: We corrected the redundant parts in sections 4.1 and 4.2 as suggested by the referee
Comment 3: The role of PPAR-γ is described inconsistently (as both upregulated and downregulated) depending on context. While such context-dependence is plausible, it requires clearer framing to avoid confusion. A summary figure or table clarifying which effects occur under which conditions (e.g., dose, model system, tissue type) would improve clarity.
Response:
A detailed mechanism of PPARP γ upregulation and downregulation was intruduced as well as two schemes (Figure 3 and Figure 5), as requested by the referee:
„Upregulation of PPAR-γ by curcumin in the context of inflammation and ROS generation causes a heterodimerization with retinoid X receptor (RXR), the molecular assembly formed will bind to peroxisome proliferation response element (PPRE) followed by induc-tion of gene transcription and suppression of proinflammatory cytokine release [74,75]. Like any protein structure, PPAR-γ undergoes posttranslational modifications of which the most important is acetylation. PPAR-γ acetylation in BAT causes mitochondrial dys-function, decreased UCP-1 expression and whitening. Intense acetylation is associated with obesity and aging [76]. In the browning phenomenon, PPAR-γ deacetylation by sirtuin-1 (SIRT-1) is observed [77], curcumin stimulating SIRT-1 thus promoting PPAR-γ deacetylation [76,78]. Furthermore, SIRT-1 deacetylates and activates peroxisome prolif-erator-activated receptor G coactivator 1-α (PGC-1α) and promotes mitochondrial biogene-sis [75], but also reduces the translocation of dynamin-related protein 1 (DRP-1) from the cytoplasm to the mitochondria, reducing mitochondrial fragmentation and protecting mi-tochondrial integrity [79] (Figure 3).”
„ Among these factors, the main regulator of adipocyte differentiation is PPAR-γ, which occurs in two forms: PPAR-γ1 expressed in all tissues except muscle and PPAR-γ2 expressed only in adipose tissue and the intestine [75,119]. The most adipogenic form is PPAR-γ, being the form inducible by HFD, weight gain seems to be due to PPAR-γ under conditions of energy excess [77]. Activation of PPAR-γ in adipocytes determines lipogenesis with increased expression of C/EBP-α [120] as well as metabolic phenomena such as the release of free fatty acids (FFA) from lipoproteins, intracellular transport of FFA, esterification with glycerol and formation of TGL [120-122]. Reduction of PPAR-γ and C/EBP-α expressions by curcumin contributes to the reduction of TGL accumulation in adipocytes and the decrease of plasma concentrations of cholesterol and TGL [118] (Figure 5)”
Comment 4: Since there are limited human clinical data, please moderate/adjust the conclusion to better reflect the current evidence, emphasizing that although preclinical results are encouraging, larger and longer-term clinical trials are needed to substantiate efficacy in humans.
Response:
We introduced a comment in conclusin section as sugested by the referee
„In humans, the studies conducted have not accurately reflected the results observed in vitro or using lab animals. This is probably due to the complexity of the enzyme systems and the cellular regulatory and signaling pathways, but also to the limited time of the research. Consequently, conditions and formulations will have to be found in which the laboratory findings will find greater similarities in human subjects.”
Comment 5: A thorough linguistic revision/editing is necessary for academic precision.
Response: We checked the english language through a native speaker
Reviewer 2 Report
Comments and Suggestions for Authors
This is a very interesting review article about adipo-modulation by turmeric bioactive phenolic components.
Comments
- Are Figures 1a, 1b, and Figure 2 original or from other cited references?
- Correct 400C.
- Insert the chemical structure for all the compounds present in the work.
- What are the side effects of turmeric?
- What are the limitations and further prospects of the collected literature?.
- Authors should make efforts to insert some Tables with the bioactivities of turmeric and other phenolic compounds from the Curcuma plant.
- Are any works done in antimicrobial activities about phenolic compounds from the Curcuma plant?
Author Response
Comment 1: Are Figures 1a, 1b, and Figure 2 original or from other cited references?
Response: In response to this referee, all figures in the manuscript, including the graphical abstract, were drawn by C.F., A.C, and R.P using exclusively ppt tools. The articles that contributed to the drawing of these figures are cited in the text. The figures represent a schematic presentation in a personalised view of already known synthesis and interconversion pathways.
Comment 2: Correct 400C.
Response: We corrected error 400C with 400C
Comment 3 and Comment 6: Insert the chemical structure for all the compounds present in the work.; Authors should make efforts to insert some Tables with the bioactivities of turmeric and other phenolic compounds from the Curcuma plant.
Response: We introduced a table, and chemical structure of all compounds as sugested by the referee. A short paragraph introduces the table into the text: „In addition to curcumin, curcuminoids and intermediates shown in Figures 1 and 2 (p-coumarinic acid, caffeic acid, ferulic acid, calebin), there are a number of products con-sidered minor but with an important contribution to the overall bioactivity of turmeric, in general [151-154]. These components are summarized in Table 2.”
Comment 4: What are the side effects of turmeric?
Response:
We introduced a paragrapgh with side effects of turmeric as requested by the referee:
The administration of curcumin is generally free of adverse effects. Over the last few years, increased attention has been given to the possibility that turmeric, frequently consumed as a food or dietary supplement, may cause adverse effects. According to recent studies, there is evidence that combining curcuma longa with black pepper, which improves its bioavailability, can lead to liver injury [17,18]. In addition, there is some research in the literature indicating that turmeric can produce allergic reactions such as contact urticaria and contact dermatitis [19,20]. Moreover, excessive intake of turmeric supplements could enhance the risk of developing kidney stones, which is a major concern for susceptible patients [20]. Curcumin has been associated with a significant risk of bleeding among patients who are on warfarin and anticoagulant medications [21]. Other common adverse effects triggered by the use of turmeric supplement include nausea, diarrhea and flatulence [22,23]. Despite these aspects, most of the data from the scientific literature recognize turmeric as being well tolerated, and according to the Food and Drug Administration (FDA), it is considered to have a good safety profile [24-28].
Comment 5: What are the limitations and further prospects of the collected literature?.
Response: We introduced a paragraph about limitations, extended our comments related to the real contribution and limitation of clinical eficacy and literature limitations. „Despite a large number of research and review articles on the use of dietary supplements in general, their administration is not accompanied by increased efficacy, persistent effects over time and clear results demonstrating significant weight loss. Batsis J.A. et al [176] underline this in a systematic review, even though, as is known, dietary supplements, es-pecially antioxidants, have their well-defined role as adjuvants being included in some therapeutic strategies even in some types of cancer [177-180]. Other limitations of the liter-ature study that addresses research on human subjects consist in the transposition of the-oretical information and in vivo experiments on laboratory animals to humans, where the phenomena observed in controlled laboratory experimental conditions are insignificant in the population studied”.
Comment 7: Are any works done in antimicrobial activities about phenolic compounds from the Curcuma plant?
Response: We introduced a paragraph related to the antimicrobial activity as sugested bu the referee: „But microbial infections are also accompanied by inflammatory phenomena. A number of studies demonstrate the antimicrobial effect of turmeric and curcumins. Among microbial species, the antibacterial effect has been observed on some gram-positive strains, for ex-ample, Staphylococcus aureus [87-89], Enterococcus fecalis [89] but also gram-negative ones such as, E. coli [87,89], Helicobacter pylori [87], Proteus mirabilis, Pseudomonas aeruginosa [89] acid-fast, Mycobacterium tuberculosis [87] but also on Candida species [87,90]. Furthermore, there are studies that have investigated the rela-tionship between curcumin and gut microbiota (GM). Curcumin is degraded in the diges-tive tract by local bacterial flora into tetrahydrocurcumin, dihydroferulic acid, dihydro-curcumin, hexahydrocurcumin [91] but also exerts a protective effect on these microor-ganisms, e.g., Bacteroidaceae, Rikenellaceae [92,93], Bifidobacterium, Lactobacillus [93]. Thus, curcumin and curcuminoids could indirectly intervene in the amelioration of obesity, since GM can regulate monosaccharide absorption from the digestive tract and modulate hepatic lipogenesis [93,94]. It has been shown that HFD depresses beneficial GM and promotes hepatic steatosis [94]. In a study conducted on B6.V-Lep ob/ob JRj mice treated with 0.3% curcumin, it was demonstrated that turmeric promotes the multiplica-tion of Bifidobacterium spp, Lactobacillus and contributes to the reduction of hypertri-glyceridemia after 5 weeks [95]. Also, among the metabolites, tetrahydrocurcumin has an important therapeutic role by improving hepatic steatosis in obese C57/BL/6 mice [96].”
Round 2
Reviewer 2 Report
Comments and Suggestions for Authors
The authors answered all the comments and amended the article in a good way. So, I suggest accepting the article for publication.